# Childhood Outcomes in Children with and without Cardiac Echogenic Foci: An Electronic Birth Cohort Study in Wales, UK

**DOI:** 10.3390/children10071233

**Published:** 2023-07-17

**Authors:** Lisa Hurt, Orhan Uzun, Susan Morris, Jackie Bethel, Annette Evans, Michael Seaborne, Rhian Daniel, Sinead Brophy, Shantini Paranjothy

**Affiliations:** 1Division of Population Medicine, School of Medicine, Cardiff University, Neuadd Meirionnydd, Heath Park, Cardiff CF14 4YS, UK; bethelja@cardiff.ac.uk (J.B.); danielr8@cardiff.ac.uk (R.D.); 2University Hospital of Wales, Cardiff and Vale University Health Board, Heath Park Way, Cardiff CF14 4XW, UK; orhan.uzun@wales.nhs.uk (O.U.);; 3Research and Evaluation Division, Public Health Wales, Cardiff CF10 4BZ, UK; 4Swansea University Medical School, Data Science Building, Singleton Park, Swansea SA2 8PP, UK; m.j.seaborne@swansea.ac.uk (M.S.); s.brophy@swansea.ac.uk (S.B.); 5Aberdeen Centre for Health Data Science, University of Aberdeen, Polwarth Building, Foresterhill, Aberdeen AB25 2ZD, UK; shantini.paranjothy@abdn.ac.uk; 6Public Health Directorate, NHS Grampian, Summerfield House, 2 Eday Road, Aberdeen AB15 6RE, UK

**Keywords:** cardiac echogenic foci, hospital admissions, congenital cardiac anomalies, cohort study, population-based

## Abstract

There is uncertainty about outcomes associated with cardiac echogenic foci (CEF) seen at the midtrimester ultrasound scan because of limited population-based follow-up data. This can lead to unnecessary invasive testing and significant parental anxiety. We analysed data from a cohort study, The Welsh Study of Mothers and Babies, to examine whether children with CEF had more adverse outcomes during childhood compared with children without CEF. Children born between 1 January 2009 and 31 December 2011 were followed until 31 January 2018, migration out of Wales, or death. The primary outcome was cardiac hospital admissions, defined a priori by an expert steering group. Secondary outcomes included congenital cardiac anomalies, and hospital admissions for other causes. There was no evidence of an association between isolated CEF and cardiac hospital admissions (hazard ratio 0.87, 95% confidence interval [CI] 0.33–2.25, *p* value 0.768), or with congenital cardiac anomalies. There was a small increased risk of a respiratory admission with isolated CEF (hazard ratio 1.27, 95% CI 1.04–1.54, *p* value 0.020). Further research is needed on features of CEF, such as location or number, to fully understand the clinical significance of these findings.

## 1. Introduction

Cardiac defects are one of the commonest congenital abnormalities [1]. They are diagnosed in at least one in 180 births [2,3] and are an important cause of mortality and morbidity in childhood [4]. Outcomes are improved with early detection [5]. As most cases arise in low-risk populations, the midtrimester ultrasound scan provides an important prenatal screening opportunity [5]. In many countries, this scan is routinely offered to all women between 18 and 20 weeks of pregnancy [6,7], and it is estimated that around a half of all structural congenital cardiac anomalies will be detected during this scan [8].

Non-structural findings (sometimes known as “markers”) of unknown clinical significance are also identified at this scan [9]. These include cardiac echogenic foci (CEF), which are echogenic areas on the papillary muscle on either or both of the atrioventricular valves. Pathological studies [10,11] have suggested that these may be areas of microcalcification or fibrosis. Their cause is unknown, but possible explanations include ischaemic changes as a result of abnormalities in the development of the cardiac microvasculature [10] or an inflammatory process [12]. Their prevalence at the second trimester scan is estimated at between 0.5% and 4.9% [13,14,15,16], with the wide range thought to result from differences in study populations and marker definitions.

In common with other markers, such as echogenic bowel, interest in CEF originally focused on whether they would help to diagnose aneuploidy in the fetus. Whilst there is an association between the presence of CEF and a later diagnosis of Down syndrome [17,18], it has been argued that this finding adds little to other prenatal screening tests now available [19,20]. It is unclear whether CEF can be used as a screening tool for cardiac diseases, because research examining whether they are predictive of cardiac dysfunction and/or anomalies has shown mixed results. Studies comparing detailed cardiac investigations in fetuses with and without CEF have found no significant differences in cardiac function overall [21,22,23,24]. However, mild impairment in diastolic functioning was found in one of these studies [24], leading to calls for further research on these foci [12].

Doubt remains about the clinical outcomes associated with this ultrasonographic finding because there have been few studies that have compared outcomes in children with and without CEF in a low-risk population. This uncertainty has the potential to lead to inappropriate referrals (for example, to genetic counselling services), unnecessary invasive testing (such as amniocentesis), and significant anxiety for families [25]. We conducted a prospective, population-based cohort study to compare outcomes in children with and without ultrasound findings of unknown significance detected at the midtrimester or fetal anomaly scan (FAS). It includes longer follow-up than any of the previous studies of CEF. In this analysis, the aim was to examine whether children with CEF had more adverse outcomes during childhood compared to children without CEF.

## 2. Materials and Methods

The Welsh Study of Mothers and Babies was a cohort study that was set up to examine the longer-term health outcomes associated with non-structural findings (including CEF) at the FAS in a cohort of pregnant women receiving routine antenatal care in Wales [26]. Ethical approval for the original study was granted by the Multicentre Research Ethics Committee for Wales (reference 08/MRE09/17) on 16 April 2008. The Methods and Results are reported as per the guideline for Strengthening the Reporting of Observational Studies in Epidemiology (STROBE [27], see S1 STROBE Checklist).

### 2.1. Study Population

All pregnant women were eligible for inclusion if they had a singleton pregnancy and attended for a second trimester FAS in six of seven Welsh Health Boards. Recruitment took place with staggered start dates from July 2008 and continued until March 2011. As part of the recruitment process, participants were asked to give written consent that the data from their ultrasound scan could be collected and linked with routinely collected data on their child. Follow-up was from birth until 31 January 2018 (end of follow-up), migration out of Wales, or death.

### 2.2. Definition of Exposure

CEF were defined as the presence of echogenic areas (as bright as bone) on the papillary muscle of either or both of the atrioventricular valves at the 18–20-week ultrasound scan. Scan data were captured using an additional reporting screen within the information system for radiological data storage and reporting in Wales (Radiology Information Service 2, RadIS2). At the end of recruitment, we contacted all Health Boards to acquire missing scan data for women who had consented to take part in the study. Where possible, their scan data were downloaded from the Health Boards’ routine reporting systems.

Ultrasound scan images where a non-structural finding had been reported were reviewed by an expert Quality Assurance (QA) panel, to validate that these fulfilled the study definition. There were 858 instances of CEF reported in the original data collection (bootstrapped prevalence of 44.9 per 1000 singleton pregnancies, accounting for missing data). Of these scans, 702 (81.8%) were reviewed by the QA panel, and the presence of CEF was confirmed in 615 (bootstrapped prevalence of 43.7 per 1000 singleton pregnancies; for more detail, see [16]).

### 2.3. Data Linkage and Outcome Definitions

Data linkage was performed in the Secure Anonymised Information Linkage (SAIL) Databank [28,29], with approval for the analysis obtained from their Information Governance Review Panel. The ultrasound data were exported to SAIL to enable linkage with data on: hospital admissions in the Patient Episode Database for Wales (PEDW); all congenital anomalies from the Congenital Anomaly Register for Wales (CARIS); deaths (from the Office for National Statistics Annual District Death Extract); and migration (from the Welsh Demographic Service data). For each of these datasets, individuals were assigned a unique identifier provided by the NHS Wales Informatics Service. The linkage system uses a combination of deterministic (based on NHS numbers) and probabilistic record linkage (based on first name, surname, date of birth, gender, and phonex and soundex version of names); this linkage is more than 99.85% accurate [29]. Second-stage encryption is used by the databank before storing data, and third-stage encryption is used to create project-specific linked datasets.

The primary outcome for this analysis was a hospital admission with a cardiac cause identified in any coding position in PEDW, in the period from birth to the end of the follow-up or censoring. An admission was defined as a stay of at least one night using a hospital bed provided by the NHS in Wales under one or more consultants, and included transfers between hospitals. A list of condition codes, based on the International Statistical Classification of Diseases and Related Health Problems 10th Revision (ICD-10, [30]), for which CEF could be considered a possible marker was agreed a priori by the study steering group (see Appendix A). This group included a consultant paediatric cardiologist, a consultant radiologist, and a patient and public involvement group. The list included congenital cardiac anomalies, cardiac arrhythmias, and malignant or benign neoplasms of the heart. Admissions as a day case for postnatal investigations alone are not a part of this dataset, and these admissions would not, therefore, have been included.

Secondary outcomes were also specified a priori and defined as follows. A diagnosis of a congenital cardiac anomaly was identified as a record in CARIS or PEDW with an ICD-10 code of Q20 to Q28 (congenital malformations of the circulatory system). We also identified children with a diagnosis of any congenital anomaly, using any code from the ICD-10 Q chapter (congenital malformations, deformations, and chromosomal abnormalities) in CARIS or PEDW records. A diagnosis of Down syndrome was identified using records in CARIS or PEDW with code Q90. We also identified hospital admissions for different causes, to examine associations with admissions for congenital anomalies (admissions with a cardiac congenital anomaly and, separately, admissions with any congenital anomaly), hospitalisations with other causes that may indicate that children with CEF were generally more unwell (admissions with a respiratory illness), and admissions with causes that may be linked to CEF (specifically, admissions with all neoplasms, and separated by whether the neoplasm was malignant or benign). The ICD-10 codes used to identify these outcomes are show in Appendix A.

### 2.4. Statistical Analysis: Inclusion and Exclusion Criteria

The population for this analysis was live-born singleton children whose date of birth was between 1 January 2009 and 31 December 2011, whose mothers had consented to take part in the study, for whom the ultrasound scan data had been collected using the study data collection tool, and for whom marker data validated by the QA panel were available (see Figure 1). Pregnancies that ended in a stillbirth or a spontaneous or induced loss were excluded, as were pregnancies with an unknown outcome (for example, because the birth happened outside of Wales). If children could not be assigned with an anonymised linking field (for example, because they did not access their healthcare in Wales or did not have a valid NHS number or other identification variables), they were also excluded because linkage with the healthcare datasets was not possible. Follow-up was from birth until the 31 January 2018 (end of follow-up), migration out of Wales, or death. Person-time was censored in cases of migration or death.

### 2.5. Statistical Analysis: Power Calculations

Preliminary data from PEDW between 1990 and 2015, examined when the study was being planned, suggested that there was a 0.75% risk of a cardiac admission (using ICD-10 codes I00 to I99, P29 and Q20–Q28) before a child’s fifth birthday. Given this estimated risk and the number of children with CEF, the sample size available for analysis was calculated to be adequate to detect a three-fold increase in the risk of cardiac admissions with 80% power. Several of the secondary outcomes (such as all hospitalisations and hospitalisations for respiratory causes) were known to be more common. Therefore, a smaller effect size would be detectable for these outcomes at the same sample size and power.

### 2.6. Statistical Analysis: Methods

A Cox proportional hazards regression model was used to model time to the first cardiac hospital admission. Fifty one percent of children with a cardiac admission were admitted more than once with the same outcome during the study period. Therefore, we also used the Andersen–Gill extension of a Cox model to examine whether associations differed in the presence of recurrent cardiac admissions, where the correlation between events was captured by appropriate time-dependent covariates [31]. We estimated hazard ratios (HRs) with 95% confidence intervals (CI) to examine the risk of hospital admissions associated with the presence of CEF at the FAS. Results from both models were similar, and we present the estimates from both in this paper for comparison. The proportional hazards assumption was assessed graphically using log-minus-log plots and was tested based on the Schoenfeld residuals. We examined associations in unadjusted models and conditional on other predictors of hospital admissions (sex, maternal age in three categories (<25, 25–34, 35+ years), deprivation quintile based on the UK Townsend Deprivation Score [32], mode of delivery, and prematurity). There was also a low percentage of children with missing data on co-variates (0.8% for Townsend score, 0.9% for mode of delivery, 0.2% for gestational age; see Table 1). Multiple imputation with chained equations [33] was used to impute values for the missing data (10 imputations) under the missing at random assumption, with parameter estimates and their standard errors combined using Rubin’s rules. The imputation model included all co-variates, the outcome variable (with different imputations conducted for each different outcome), and the cumulative baseline hazard [34]. Conclusions from a complete case analysis and following multiple imputation were similar, and we present the results from the analysis using multiple imputation in this paper. All analyses were conducted within the SAIL Gateway using Stata version 16 [35]. The SAIL Databank uses small number suppression to ensure that no individuals can be re-identified from data presented in publications. No data can therefore be presented for cells with fewer than five cases.

## 3. Results

A total of 22,045 children with anomaly scan data were eligible for inclusion in the study (Figure 1). A total of 18,246 pregnancies (83% of pregnancies with anomaly scan data) were eligible for inclusion in this analysis. Pregnancies were excluded if: scan data were only available from routine reports (therefore did not include data on CEF, *n* = 2920); the scan images were not available for QA (*n* = 252); the pregnancy did not end in a live birth (stillbirths *n* = 64, spontaneous or induced pregnancy loss *n* = 37); pregnancy outcome data were not available (*n* = 504); or a unique identifier could not be assigned to the infant within the SAIL Databank (therefore, data linkage with hospital admissions was not possible, *n* = 22). The characteristics of the included mothers and their pregnancy outcomes were comparable to the general population of pregnant women in Wales (Table 1, see [16]). Sixty one of the children died during follow-up (median age at death 19 days (IQR 5, 160)). A total of 786 children moved out of Wales (median age at move 3.19 years (IQR 1.4, 5.1)). The median follow-up time for the cohort was 7.32 years (IQR 6.8, 7.8).

A total of 596 children in this sample had confirmed CEF at the FAS (Table 1). For 585 children, this was an isolated finding. Of the 11 children with multiple markers, the commonest co-occurring marker was renal pelvis dilatation (specific numbers cannot be reported as *n* < 5). CEF were more prevalent in children of younger mothers (4.0% when maternal age was <25), with area-level social deprivation (4.1% in the most deprived area), and in children born preterm (4.5%), but there was no association with sex of the child or mode of delivery (vaginal compared with Caesarean section). Fewer than five cases of Down syndrome were identified in the cohort. Information on the characteristics of these children cannot therefore be presented.

Of the 324 children with a cardiac admission, 51.0% had multiple admissions (total cardiac admissions = 661). There was no evidence of an association between the presence of CEF on the FAS and a hospital admission with a cardiac cause (Table 2). Patterns were similar in univariate analyses, when time to first admission was examined (cHR 0.82, 95% CI 0.44, 1.55, *p* value 0.547), or when estimates were adjusted to account for multiple admissions (cHR 0.87, 95% CI 0.33, 2.25, *p* value 0.768). Data could not be examined separately for children with isolated CEF or CEF with another marker, as there were no cardiac admissions in the latter group. In addition, data could not be analysed by individual cardiac causes (for example, cardiac arrhythmias or cardiomyopathy) as there were too few admissions with these codes in the cohort.

Ten of the children with CEF (4.2%, all with isolated CEF) had a congenital cardiac anomaly, compared with 586 (3.3%) of the children without CEF at the FAS. There was no evidence of an association between the presence of CEF on the FAS and congenital cardiac anomalies in the univariate or adjusted models, or when hospital admissions with a congenital cardiac anomaly code were examined (Table 3). Thirty eight of the children with CEF (3.7%) had a congenital anomaly (any Q code), compared with 558 (3.2%) of the children without CEF at the FAS. There was no evidence of an association between the presence of CEF on the FAS and any congenital anomalies when all children with CEF were examined as one group, but children with CEF and another marker were five times as likely to have a congenital anomaly as children without any markers at the FAS (cOR 5.03, 95% CI 1.26, 20.10, *p* value 0.022). This association was not replicated in the analysis of hospital admissions with any congenital anomaly code (Table 3).

Table 4 shows the association between the presence of CEF on the FAS and hospital admissions with respiratory or neoplasm codes. Children with CEF had a small increased risk of an admission with a respiratory cause in the adjusted analyses once multiple admissions were accounted for (cHR 1.27, 95% CI 1.04, 1.54, *p* value 0.020), with the increase seen only in children with isolated CEF. There was no evidence of an association between CEF and hospitalisations for any neoplasms (benign or malignant), with no cases on malignant neoplasms in the CEF group. No cases of rhabdomyoma were identified in the whole cohort.

## 4. Discussion

In this population-based cohort, there were 596 singleton pregnancies in which CEF was identified at the fetal anomaly ultrasound scan. There was no evidence of an association between a finding of CEF at the scan and our primary outcome, hospital admissions for cardiac causes. There was also no evidence of an association between CEF and congenital cardiac anomalies. Children with CEF and another marker were more likely to be diagnosed with any congenital anomaly compared with children without CEF. Children with isolated CEF had an increased risk of multiple hospital admissions for respiratory causes, but this was small and it is unclear from these data whether this is clinically significant.

These results are consistent with previous studies that have suggested no association between CEF and congenital cardiac anomalies [36] or chromosomal abnormalities [37]. Previous studies have shown that there is an association between multiple markers and adverse outcomes, for example, Hu et al. [37] who found more chromosomal abnormalities in fetuses with these findings. Our findings are consistent with guidance for practice (for example, from the American College of Obstetrics and Gynecology, [38]), which recommend additional investigations, genetic counselling, and maternal–fetal medicine consultation when more than one marker is identified at the FAS.

Our study is an important addition to the evidence on CEF because it was a large population-based study with follow-up for several years into childhood, and unlike previous studies, we could compare outcomes in children with and without CEF. We included stringent QA processes, so that we can be certain that the foci identified conform to standard definitions and are unlikely to be artefactual. Reassuringly, the prevalence of CEF in our sample was consistent with previous studies in low-risk populations [15], and suggests that the QA process did not lead to the exclusion of true foci. Data linkage with routinely collected healthcare records ensured that few participants were lost to follow-up, with data available for 97% of women and children. Data sources that cover the whole population in Wales, such as routinely available healthcare records on hospital admissions and data from a national registry of congenital anomalies, were used to capture data on outcomes in this study. The results are likely to be generalisable to populations outside of Wales with similar access to healthcare.

Overall, the total number of pregnancies that could be included in the analysis was reduced due to issues with the initial data collection, which meant that sonographers did not access the study data collection screen when conducting the scans. Although this has reduced our sample size overall, this study remains one of the largest cohorts to have examined this marker in a low-risk population, with almost 600 pregnancies with CEF included. The included cohort was also representative of all pregnant women in Wales, and it is reassuring that we have been able to rule out a strong association between this finding and cardiac hospital admissions and congenital cardiac anomalies in childhood in this population.

We were unable to obtain information about the features of the foci (such as their location, size, or number) in this study. Recent studies have suggested that the location [39] (and specifically whether the foci are in the right ventricle) or number [40] of foci may be important to predict the presence of congenital heart disease. Chiu et al. [36] also found—in a low-risk cohort—that left-sided CEF are most likely to resolve or disappear later in pregnancy, whereas right-sided findings are more likely to persist. Although we could not identify the features of the CEF in our study, our previous analysis suggested a small increased risk in preterm birth in children with CEF and, in this study, we identified a small increased risk of respiratory admissions. These findings may be indicative of an increased risk in a sub-group of children that we have not been able to identify. Therefore, further research is needed to understand whether foci with specific features are indicators of adverse outcomes, and to inform decisions about further follow-up and the communication of these risks to families. However, conducting large population-based cohort studies is expensive and time consuming, and even these may lack the sample size to stratify outcomes according to different CEF features. An important first step may be to conduct a systematic review and meta-analysis of existing studies, with the analyses stratified by CEF features and the risk profile of the pregnancies. Following this, informed decisions could be made about whether it may be appropriate to include data collection on CEF within routine health service systems to monitor outcomes on a population basis.

Doubt remains about this ultrasound marker because its origin remains unknown, and previous studies have found conflicting results. Our study adds to the knowledge by demonstrating that there is no evidence of an increase in hospital admissions for cardiac causes, congenital cardiac anomalies, or any congenital anomaly in children with isolated CEF in this population-based cohort. Further research is needed on the risks associated with different features of CEF, such as location or number.

## Figures and Tables

**Figure 1 children-10-01233-f001:**
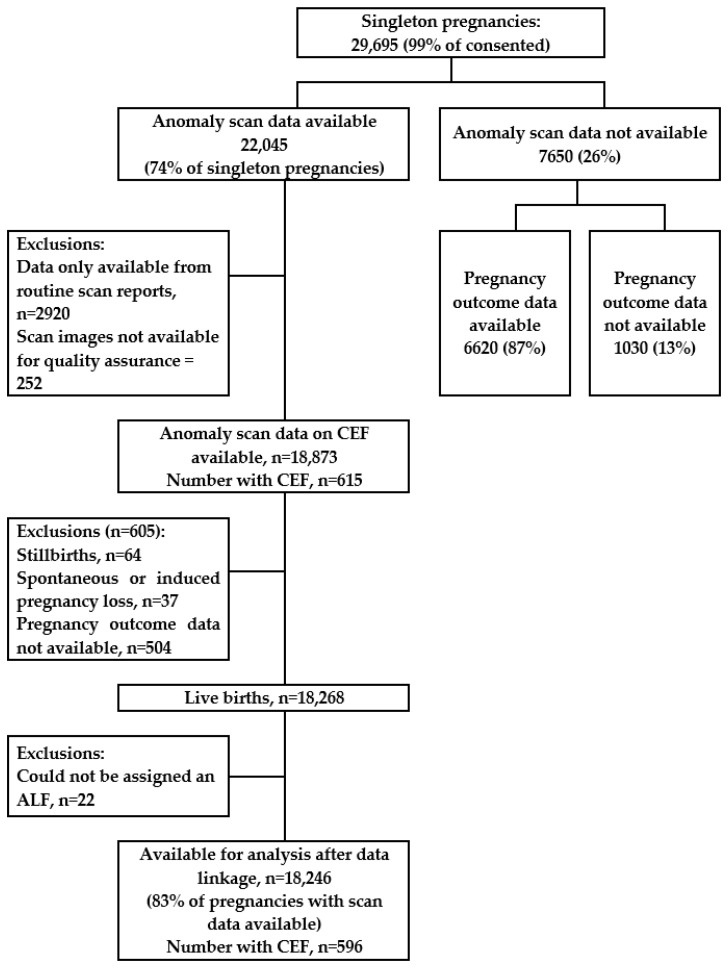
Cohort flow diagram.

**Table 1 children-10-01233-t001:** Characteristics of the cohort.

	**Total**	**Cardiac Echogenic Foci** **at the Fetal Anomaly Scan**	
		**No**	**Yes**
	**N**	**N (%)**	**N (%)**
	**18,246**	**17,650 (96.7)**	**596 (3.3)**
**Sex**			
Female	8845 (48.5%)	8559 (48.5)	286 (48.0)
Male	9401 (51.5%)	9091 (51.5)	310 (52.0)
**Maternal age (years)**			
<25	5384 (29.5%)	5169 (29.3)	215 (36.0)
25–34	10,075 (55.2%)	9779 (55.4)	296 (49.7)
35+	2787 (15.3%)	2702 (15.3)	85 (14.3)
**Townsend Score ***			
1	2962 (16.4%)	2878 (16.4)	84 (14.1)
2	2964 (16.4%)	2880 (16.5)	84 (14.1)
3	3586 (19.8%)	3466 (19.8)	120 (20.1)
4	4248 (23.5%)	4117 (23.5)	131 (22.0)
5	4342 (24.0%)	4165 (23.8)	177 (29.7)
**Birth by Caesarean Section ****			
Yes	4517 (25.0%)	4383 (25.1)	134 (22.5)
No	13,558 (75.0%)	13,097 (74.9)	461 (77.5)
**Preterm birth †**			
Yes	966 (5.3%)	922 (5.2)	44 (7.4)
No	17,237 (94.7%)	16,687 (94.8)	550 (92.6)

* Townsend deprivation score: 1 = least deprived, 5 = most deprived; 144 pregnancies (none with CEF) with missing data for this variable. ** 171 pregnancies (<5 with CEF) with missing data for C-Section. † Preterm birth = <37 weeks gestation; 43 pregnancies (<5 with CEF) with missing data for gestational age; Total N with missing data in any variables = 220.

**Table 2 children-10-01233-t002:** Association between CEF and cardiac hospital admissions.

	Conditional HR, Any Admission * (95% CI)	*p* Value	Conditional HR, Multiple Admissions ** (95% CI)	*p* Value
Hospital admissions with all cardiac codes				
No CEF	1.00		1.00	
CEF	0.82 (0.44, 1.55)	0.547	0.87 (0.33, 2.25)	0.768
No CEF	No cardiac admissions in multiple marker group			
Isolated CEF				
CEF with another marker				

HR = hazard ratio. * First admission only (*n* = 324), conditional on sex, maternal age, Townsend score, preterm birth and C-section; and all results from the analysis using multiple imputation. ** Estimate also adjusted for multiple admissions using Anderson–Gill model (total number of admissions = 661).

**Table 3 children-10-01233-t003:** Association between CEF and presence of congenital cardiac anomalies and all congenital anomalies, overall and in hospital admission records.

**Congenital Cardiac Anomaly and All Congenital Anomaly Cases**				
	**Univariate OR** **(95% CI)**	***p* Value**	**Conditional OR** **(95% CI) ***	***p* Value**
**Congenital cardiac anomalies**				
No CEF	1.00		1.00	
CEF	1.32 (0.69, 2.49)	0.400	1.19 (0.62, 2.28)	0.593
No CEF	No congenital cardiac anomalies in the multiple marker group			
Isolated CEF				
CEF with another marker				
**Any congenital anomalies**				
No CEF	1.00		1.00	
CEF	1.14 (0.82, 1.59)	0.440	1.08 (0.77, 1.51)	0.671
No CEF	Cannot be presented because *n* < 5 in some cells		1.00	
Isolated CEF			1.01 (0.71, 1.44)	0.958
CEF with another marker **			5.03 (1.26, 20.10)	0.022
**Hospital admissions with congenital cardiac anomaly and all congenital anomaly codes**				
	**Conditional HR,**	***p* value**	**Conditional HR, multiple admissions ^††^** **(95% CI)**	***p* value**
	**any admission ^†^**			
	**(95% CI)**			
**Hospital admissions with congenital cardiac anomaly codes ^‡^**				
No CEF	1.00		1.00	
CEF	1.06 (0.50, 2.26)	0.886	1.16 (0.38, 3.52)	0.793
No CEF	No congenital cardiac admissions in multiple marker group			
Isolated CEF				
CEF with another marker				
**Hospital admissions with any congenital anomaly codes ^‡‡^**				
No CEF	1.00		1.00	
CEF	1.13 (0.80, 1.60)	0.491	0.93 (0.55, 1.58)	0.803
No CEF	1.00		1.00	
Isolated CEF	1.08 (0.76, 1.55)	0.661	0.90 (0.52, 1.56)	0.707
CEF with another marker	3.39 (0.84, 13.61)	0.085	2.63 (0.72, 9.65)	0.143

OR = odds ratio. HR = hazard ratio. * Conditional on sex, maternal age, Townsend score, preterm birth and C-section; ** commonest co-occurring marker is renal pelvis dilatation (numbers cannot be presented as *n* < 5). ^†^ First admission only, conditional on sex, maternal age, Townsend score, preterm birth and C-section; and all results from the analysis using multiple imputation. ^††^ Additionally adjusting for multiple admissions using Anderson–Gill model. ^‡^ Number of first admissions = 174; total number of admissions = 418. ^‡‡^ Number of first admissions = 825; total number of admissions = 1869.

**Table 4 children-10-01233-t004:** Association between CEF and hospital admissions with other causes.

	Conditional HR, Any Admission * (95% CI)	*p* Value	Conditional HR, Multiple Admissions ** (95% CI)	*p* Value
Hospital admissions with a code for respiratory illnesses ^†^				
No CEF	1.00		1.00	
CEF	1.08 (0.93, 1.26)	0.325	1.27 (1.04, 1.54)	0.020
No CEF	1.00		1.00	
Isolated CEF	1.10 (0.94, 1.28)	0.244	1.27 (1.05, 1.56)	0.015
CEF with another marker	0.31 (0.04, 2.23)	0.248	0.56 (0.10, 3.22)	0.520
**Hospital admissions with a code for any neoplasms (benign and malignant) ^††^ **				
No CEF	1.00		1.00	
CEF	0.60 (0.15, 2.43)	0.472	0.32 (0.05, 1.98)	0.220
No CEF	No neoplasm admissions in multiple marker group			
Isolated CEF				
CEF with another marker				
**Hospital admissions with a code for benign neoplasms ^‡^**				
No CEF	1.00		1.00	
CEF	0.70 (0.17, 2.85)	0.619	1.15 (0.17, 7.79)	0.885
No CEF	No neoplasm admissions in multiple marker group			
Isolated CEF				
CEF with another marker				

HR = hazard ratio. * First admission only, conditional on sex, maternal age, Townsend score, preterm birth and C-section; and all results from the analysis using multiple imputation. ** Additionally adjusting for multiple admissions using Anderson–Gill model. ^†^ Number of first admissions = 4721, total number of admissions = 7749. ^††^ Number of first admissions = 19; total number of admissions = 458; no malignant neoplasm admissions in the CEF group. ^‡^ Number of first admissions = 86; total number of admissions = 191.

## Data Availability

The data used in this study are available in the SAIL databank at Swansea University, Swansea, UK. SAIL has established an application process to be followed by anyone who would like to access data via SAIL, https://www.saildatabank.com/application-process. All proposals to use SAIL data are subject to review by an independent Information Governance Review Panel (IGRP). Before any data can be accessed, approval must be given by the IGRP. The IGRP gives careful consideration to each project to ensure proper and appropriate use of SAIL data. When access has been granted, it is gained through a privacy-protecting safe haven and remote access system referred to as the SAIL Gateway. Relevant information to allow acquisition of a replicable data set is available in the paper and its Supporting Information files, or can be requested from the authors. Please contact SAILDatabank@swansea.ac.uk for more detail on data access requests.

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
