# Peer review of "Childhood Outcomes in Children with and without Cardiac Echogenic Foci: An Electronic Birth Cohort Study in Wales, UK"

_children, 2023, doi:10.3390/children10071233_

Round 1
Reviewer 1 Report
In this study the authors sought to evaluate the association between a prenatal diagnosis of cardiac echogenic foci (CEF) and hospital admissions for cardiac causes and also congenital cardiac anomalies or others diseases. They didn’t found any significant association between isolated CEF and cardiac hospital admissions or with congenital cardiac anomalies. However children with CEF and another marker were more likely to be diagnosed with any congenital anomaly compared with children without CEF, and children with isolated CEF had a small increased risk of multiple hospital admissions for respiratory causes.
Some comments have to be addressed:
-Abstract The authors reported: “children were followed from birth until 31st January 2018 could the authors report that the study include children born between Jan 1, 2009, and Dec 31, 2011”
- Introduction
- Perhaps too long a part could be moved to the discussion
- The authors reported “However, mild impairment in diastolic functioning was found in one of these studies [21], I haven’t access to the full text, but in the abstract the article cited concluded that An isolated IEF is not associated with abnormal cardiac function. Please check
- Could the authors explain why they report the associations between participant characteristics and cardiac admissions, congenital cardiac anomalies, and any congenital anomaly is beyond the aim of the study?
- but children with CEF and another marker were five times as likely to have a congenital anomaly as children without any markers could the authors report which markers
- The authors reported “children with CEF had a small increased likelihood of an admission with a respiratory cause once multiple admissions this was adjusted or not adjusted analysis
Table 1 could the author reporter the percentage of the variables in each group for instance the percentage of male/female.. and the significance between the groups.
Author Response
Please see attached document. As some of the comments from both reviewers were similar, we have included our response to both reviewers within this document.

Reviewer 2 Report
Thank you for conducting a great study
in table 1 the (%) ought to be relevant to the n of the first row in the that column. Currently it is confusing as it suggest way more prevelance in the group without CEF
table 2 and all that paragraph can be deleted. Please focus the entire manuscript in CEF versus no-CEF
Good job
Author Response

(The authors gave the same response as above.)
